# Neglected Tropical Diseases

# Shotgun metagenomic analysis of the oral microbiomes of children with noma

Michael Olaleye[1], Angus M. O'Ferrall[2,3], Richard N. Goodman[2,3], Deogracia Wa Kabila[1], Miriam Peters[1], Gregoire Falq[4], Joseph Samuel[1], Donal Doyle[5], Diana Gomez[5], Gbemisola Oloruntuyi[1], Shafi'u Isah[6], Adeniyi Semiyu Adetunji[6], Elise Farley[1], Nicholas. J. Evans[7], Mark Sherlock[5], Adam P. Roberts[2,3]*, Mohana Amirtharajah[5]*, Stuart Ainsworth[7]*

1 Noma Children's Hospital, Médecins Sans Frontières, Sokoto, Nigeria, 2 Department of Tropical Disease Biology, Liverpool School of Tropical Medicine, Liverpool, United Kingdom, 3 Centre for Neglected Tropical Diseases, Liverpool School of Tropical Medicine, Liverpool, United Kingdom, 4 Manson Unit, Médecins Sans Frontières, London, United Kingdom, 5 Operational Centre Amsterdam, Médecins Sans Frontières, Amsterdam, The Netherlands, 6 Department of Clinical Services, Noma Children's Hospital, Sokoto, Nigeria, 7 Department of Infection Biology and Microbiomes, Institute of Infection, Veterinary and Ecological Sciences, University of Liverpool, Liverpool, United Kingdom

☉ These authors contributed equally
* adam.roberts@lstmed.ac.uk (APR), mohana.amirtharajah@amsterdam.msf.org (MA), stuart.ainsworth@liverpool.ac.uk (SA)

## Abstract

Noma is a rapidly progressive orofacial gangrene that predominantly affects children living in extreme poverty. Despite its documentation since antiquity and its designation as a World Health Organisation Neglected Tropical Disease in 2023, the microbiological cause of noma remains poorly understood, with no specific organisms confidently identified as definitive aetiological agents. Here, we present the first deep shotgun metagenomic profiling of oral saliva microbiomes from 19 Nigerian children with acute noma. Our analyses of this preliminary study reveal marked microbial dysbiosis in noma microbiomes, with machine learning and multivariate statistical analyses indicating significant enrichment of *Treponema*, *Porphyromonas*, and *Bacteroides*, alongside depletion of *Streptococcus* and *Rothia*, as key microbial signatures of noma disease. From the dataset we recovered 40 high-quality *Treponema* metagenome assembled genomes (MAGs) spanning 19 species, 14 of which were novel. Notably, a novel species designated *Treponema* sp. A was detected in 15 of the 19 noma participants and was entirely absent from an internationally representative set of healthy saliva metagenomes. Re-analysis of previously published 16S rRNA datasets from children with noma in Niger also revealed *Treponema* sp. A to be highly prevalent in noma cases but extremely rare in controls. While these findings highlight *Treponema*, particularly *Treponema* sp. A, as an organism of interest and a potential contributor to noma pathogenesis, further comprehensive studies will be required to confirm this association and to clarify whether it reflects a causal role and/or is a genuine marker of noma dysbiosis. Additionally, analysis of antimicrobial

**Data availability statement:** Data availability All anonymized data used in this study is available in supplementary materials. Short reads from all samples sequenced in this study have been deposited in the Sequence Read Archive (SRA) under BioProject PRJNA1273078, with accession numbers SRR33858003– SRR33858030. All accessions for MAGs defined in this project can be found in S1 Data. Code availability statement All code to reproduce this analysis is publicly available through GitHub: https://rngoodman.github.io/noma-metagenomics.

**Funding:** This study was funded by Médecins Sans Frontières (MSF). M.O., D.W.K., M.P., G.F., J.S., D.D., G.O., E.F., M.S., and M.A. are employees of MSF and contributed to the study design, data collection, analysis, and manuscript preparation as members of the author team. A.M.O. is supported by the Medical Research Council via the Liverpool School of Tropical Medicine-Lancaster University doctoral training partnership (grant no. MR/W007037/1). R.N.G and A.P.R. are supported by funding from the Medical Research Council (MRC), Biotechnology and Biological Sciences Research Council (BBSRC) and Natural Environmental Research Council (NERC), which are all Councils of UK Research and Innovation (Grant no. MR/W030578/1) under the umbrella of the JPIAMR - Joint Programming Initiative on Antimicrobial Resistance via the STRESST project, and from UKRI through the Strength in Places Fund (grant no. SIPF 36348), as part of the infection innovation Consortium (iiCON). The funders had no role in study design, data collection and analysis, decision to publish, or preparation of the manuscript.

**Competing interests:** The authors have declared that no competing interests exist.

resistance determinants detected in noma metagenomes revealed concerning levels of resistance to antibiotics commonly used in noma treatment, particularly β-lactams and metronidazole, especially among *Prevotella* spp. These findings provide the first high-resolution microbial framework for noma and offer a foundation for future research into its pathogenesis and the development of novel diagnostics, therapeutics, and preventive strategies in endemic settings.

## Author summary

Noma is a rapidly progressing necrotic disease that destroys the tissues of the mouth and face, mainly affecting children living in abject poverty. Although antibiotic treatment can be lifesaving, the microbes involved in noma are still largely unknown, making prevention and early diagnosis difficult. To help address this, we used metagenomic DNA sequencing to profile bacteria from the mouths of 19 Nigerian children with noma treated at the Noma Children's Hospital in Sokoto, and compared these results with publicly available datasets from healthy individuals. The oral microbiomes of children with noma were remarkably different from healthy controls, with higher levels of certain bacterial species including *Treponema*, *Porphyromonas*, and *Bacteroides* and lower levels of bacteria commonly seen in healthy oral cavities, including *Streptococcus* and *Rothia*. We also identified a previously undescribed *Treponema* species in oral microbiomes of children with noma in this study and found it was also common in earlier noma samples from an independent study in Niger. Finally, we detected genes associated with resistance to antibiotics commonly used to treat noma, including beta-lactams and metronidazole. These findings provide a foundation for larger studies which will hopefully one day lead to diagnostics, targeted therapies, and preventive strategies for noma in endemic settings.

## Introduction

Noma (*cancrum oris*) is an orofacial gangrene of unknown etiology that predominantly affects children (most often aged 2–6 years) living in chronic poverty with limited access to healthcare [1]. Once initiated, noma rapidly destroys the soft tissues of the face, including the cheeks, lips, nose and occasionally the eyes [2]. It is estimated that, without treatment, noma case-fatality can be as high as ~90%; however, early access to appropriate care is associated with markedly lower mortality [3]. Survivors are often left with permanent debilitating scars and sequalae often requiring complex surgical intervention [4,5]. The social and economic impact of noma on impoverished communities whilst not thoroughly documented, is thought to be substantial [6] resulting in noma being declared a Neglected Tropical Disease by World Health Organization (WHO) in 2023 [7].

Determining global incidence of noma has been notoriously difficult due to lack of data, with the most widely cited figure suggesting 140,000 children a year may suffer from this severe disease [8], although the veracity of this figure has been questioned [9]. The vast majority of reported cases are from West and East Africa, particularly Nigeria as there is a large research presence there [10–12]. However, noma has a global distribution, appearing in many countries with high levels of chronic malnourishment and poor healthcare, or in immunocompromised individuals [9]. Indeed, historically, noma was reported in Europe and North America during the 19th century [13], was referred to as "trench mouth" during the first world war [14], and was routinely observed among inmates of the Nazi concentration camps of the second world war [15].

Under the current WHO classification of noma progression, introduced in 2016 [8], the disease initiates as acute necrotizing ulcerative gingival lesion (stage 1), before progressing through oedema (stage 2), gangrene (stage 3), scarring (stage 4), and sequalae (stage 5). Prompt intervention in the early stages of noma, including aggressive antibiotic treatment, nutritional supplementation, and surgical debridement, can lead to substantial reduction in mortality, albeit often still with significant sequalae and life-long socio-economic implications.

Although the major risk factors of noma are well understood and include chronic malnourishment and immunosuppression [3,16–18], the mechanism of initiation of infection, the associated microorganisms, and the role of the host immune response, are poorly understood or unknown [19]. No single etiological agent has been identified as the cause of noma, although its susceptibility to aggressive antibiotic treatment certainly suggests it is bacterial in nature. It has long been suspected that noma occurs, similar to other periodontal diseases, due to a dysbiosis in normal healthy functional oral microbiome and dysregulation or dysfunction in immune responses, driven by living in abject poverty [19,20].

Despite no single etiological agent being identified, microbiological surveys have suggested bacterial taxa which may be involved in noma pathology. A recent systematic review of published evidence on noma identified 121 taxa identified as being associated, at some point, with noma [3]. Organisms have primarily been associated with noma through culture or, to a lesser extent, 16S rRNA sequencing [3,16,19–26]. Studies focused on the microbiomes of noma participants reported substantially different microbiomes to controls, frequently identifying *Prevotella intermedia,* various spirochetes and *Fusobacterium nucleatum* as indicators of noma microbiota. However, these studies have noted technical limitations in regard to sampling sites (e.g., sampling dead tissue) and methodological biases (inherent biased amplification 16S rRNA genes, partial sequencing, and discriminatory culture media), which are acknowledged as not being able to provide an unbiased appraisal of a microbiome [19,27,28], thus not enabling definitive descriptions of noma related microbiomes.

Shotgun metagenomics overcomes many of the limitations of previous microbiological survey techniques employed for noma. The technique sequences all DNA present within a sample, thus avoiding any amplification or primer biases, providing an unbiased reflection of a metagenome [29]. Additionally, It can also offer insights into the functional potential of microorganisms in the sample, allowing retrieval of whole genomes and analysis of specific traits, such as antimicrobial resistance [30]. As far as we are aware, shotgun metagenomic sequencing has yet to be utilised in noma research [19].

Here we present a preliminary investigation on the first microbiological profiling of noma using deep shotgun metagenomics, describing the oral metagenomes of 19 participants with acute noma and retrospectively analysing existing data sets to enable us to profile the microbiological taxa we found associated with the disease.

## Results

During the recruitment period, 53 Stage 1–4 noma participants aged two to 12 years were admitted at the Noma Children's Hospital. No stage 0 participants at risk of noma were admitted during the study period. We enrolled 21 of these participants, however one participant was unable to provide a sample after enrolment, so was excluded from the study, leaving us with a total of 20 participants, while another participant's samples failed extraction (see "Total DNA extraction and metagenome sequencing" below), leaving a total of 19 enrolled participants in the study. Of the 32 participants not enrolled in the study, 18 had recently taken antibiotics, eight had been hospitalized in different health facilities before admission, five were missed due to arriving out of hours, and one had wound dressing before admission at the hospital (S1 Table).

## Sociodemographic and disease progression characteristics of enrolled participants

Amongst the 19 enrolled participants (S1 Table), the majority (n = 16, 84%) were aged two to six years, while the remaining participants were between seven and 10 years (n = 3, 16%). The participants included 10 males (52.6%) and 9 females (47.4%).

One participant was diagnosed upon admission with Stage 1 noma (acute necrotizing ulcerative gingivitis, 5.3%), 9 with Stage 2 (odema, 47%), seven with Stage 3 (gangrene, 37%), and two with Stage 4 (scarring, 11%) (S1 Table). All stage 2 and 3 participants were admitted for treatment with antibiotics and wound dressing, while the only stage 1 participant was an outpatient who received treatment but did not have anthropometric measurements recorded. All stage 1–3 participants recovered from their initial infection after receiving treatment. The two stage 4 participants were admitted for physiotherapy.

The majority of participants were from Sokoto State (n = 9, 47.4%), with the remainder coming from neighbouring states of Zamfara (n = 6, 31.6%) and Kebbi (n = 2, 10.5%), whist a single participant each from more distant Kaduna and Kano states was enrolled (S2 Table). The Noma Children's Hospital was the first healthcare facility visited by most participants (n = 15, 78.9%) (S3 Table). Of the four who had visited a health facility/ provider before visiting the Noma Children's Hospital, three (75%) had visited a primary health facility and the other a chemist/drugstore. None of these four participants were provided with any treatment at the facilities before they were referred to the Noma Children's Hospital. Amongst the 19 participants, two (10.5%) experienced challenges in seeking healthcare at the Noma Children's Hospital, both citing expensive transportation as an issue (S3 Table). Two participants (10.5%) had visited an oral health care provider for a check-up in the past year, both at primary health facilities.

Eight caregivers (42.1%) indicated that their child fell sick seven to 14 days before hospitalization at the Noma Children's Hospital, while seven caregivers (36.8%) reported that their infants became sick two to six days prior to admission (S1 Table). Participants reported several first symptoms including a painful mouth (n = 18, 94.7%), foul mouth odour (n = 14, 73.7%), red gums (n = 11, 57.9%), swollen cheek (n = 10, 52.6%) bleeding gums (n = 8, 42.1%) or a hole in their cheek (n = 6, 31.6%). Six (31.6%) of the participants had visible bleeding from the site of the wound at the time of sampling (S1 Table).

Most of the participants who were aged two to five years were not acutely malnourished, having MUAC of 12.5 cm and above, while two participants in this age bracket had moderate to severe malnutrition (S1 Table). The majority of participants (n = 19, 95%) had not previously received Haemophilus influenzae b, Human papillomavirus, Meningococcal A and C vaccines, with most (n = 17, 85%) also not having received Diphtheria, Measles, or BCG vaccines, either. The Poliovirus vaccine was an exception, with 50% coverage (n = 10).

## Total DNA extraction and metagenome sequencing

Three saliva samples and nine swab samples failed total DNA extraction, yielding insufficient DNA for sequencing. In total, noma oral metagenomes were obtained for 19 individuals (swab and saliva n = 9, saliva only n = 8, swab only n = 2). The number of raw read pairs per sample ranged from 40.23 million to 80.36 million (mean = 55.30 million), with between 2.76 and 75.56 million (mean = 12.44 million) reads remaining after trimming and host DNA removal (S4 Table). These reads had a mean Q20 of 98.73% (sd = 0.32%) and a mean Q30 of 95.52% (sd = 0.95%).

## Assessing the impact of sample type and categorical variables on noma metagenomic diversity

Initially we wished to determine whether there were any differences between swab and saliva samples collected from the same participants, as well as to assess overall differences between swab and saliva samples across the entire dataset. Comparison of the swab and saliva metagenomes obtained from the same participants (*n* = 9) demonstrates that the relative abundance of genera is similar across the two types of samples (Fig 1A) and the sample type (i.e., swab vs

saliva) was not a significant factor when using PERMANOVA ($R^2 = 0.01$, $p = 0.946$). We then used the Wilcoxon signed-rank test with Benjamini-Hochberg False Discovery Rate (FDR) correction for multiple comparisons to look at specific genera between swab and saliva samples with adjusted p-values represented as q-values. With the exception of samples from participant N9, which demonstrated a highly anomalous microbiome dominated by *Escherichia*, all other participants with both saliva and swab samples (i.e., N4, N5, N6, N7, N10, N13, N17, N18) had no significant differences between relative abundances of *Prevotella* ($q = 0.784$), *Treponema* ($q = 0.91$) *Neisseria* ($q = 0.784$), *Bacteroides* ($q = 0.784$), *Filifactor* ($q = 0.68$), *Porphyromonas* ($q = 0.808$), *Fusobacterium* ($q = 0.797$), *Escherichia* ($q = 0.784$), *Selenomonas* ($q = 0.688$), *Aggregatibacter* ($q = 0.688$) and *Capnocytophaga* ($q = 0.6875$).

### Relative abundance of top genera in noma and global healthy saliva dataset microbiomes

Twenty publicly available shotgun metagenomic saliva samples from healthy individuals from three countries (USA, Japan and Denmark) [31–33] were used to compare against the noma saliva samples from this study (S6 Table). We found significant differences between countries within the global healthy saliva dataset samples ($p = 0.001$) with anosim, mrpp, and permanova. When assessing the differences in specific genera within the global healthy saliva dataset, *Treponema* ($p = 0.001$), *Bacteroides* ($p = 0.019$), *Porphyromonas* ($p = 0.004$), *Fusobacterium* ($p = 0.001$), *Streptococcus* ($p = 0.001$), and *Rothia* ($p = 0.001$) were all significantly different between countries. However, when using differential analysis to compare the normalised counts of genera in the noma dataset against each of the healthy datasets, the noma-associated shifts in *Treponema*, *Rothia*, *Veillonella*, *Gemella*, and *Streptococcus* far exceeded the residual variation among healthy samples (S2 Fig).

Next, we compared the relative abundance of the top 22 genera between the noma and global healthy saliva dataset samples (Fig 1B) using a Wilcoxon rank-sum test, with p-values adjusted for multiple comparisons using the Benjamini-Hochberg FDR method. We also calculated the $log_2$ fold change (LFC), relative to noma samples, between the medians of the healthy and noma datasets. Of the 22 genera, 19 were significantly different between noma and global healthy saliva dataset samples ($q$-values $< 0.05$) with only *Campylobacter*, *Haemophilus,* and *Leptotrichia* not demonstrating significant differences. S3 Fig shows the differences between the top 22 genera; those more abundant in noma samples include *Treponema* (LFC = +5.79, $q = 1.38 \times 10^{-09}$), *Bacteroides* (LFC = +2.17, $q = 6.45 \times 10^{-9}$), *Filifactor* (LFC = +4.84, $q = 2.34 \times 10^{-7}$), *Porphyromonas* (LFC = +3.15, $q = 4.79 \times 10^{-7}$), *Fusobacterium* (LFC = +2.13, $q = 1.53 \times 10^{-4}$), *Escherichia* (LFC = +1.08, $q = 2.24 \times 10^{-4}$), *Selenomonas* (LFC = +2.55, $q = 3.25 \times 10^{-4}$), *Aggregatibacter* (LFC = +3.19, $q = 1.34 \times 10^{-3}$), *Neisseria* (LFC = +1.87, $q = 0.016$), *Prevotella* (LFC = +1.39, $q = 0.021$), and *Capnocytophaga* (LFC = + 1.97, $q = 0.021$). Those more abundant in the global healthy saliva dataset include *Streptococcus* (LFC = -3.41, $q = 1.38 \times 10^{-9}$), *Rothia* (LFC = -6.68, $q = 4.63 \times 10^{-8}$), *Schaalia* (LFC = -6.59, $q = 2.34 \times 10^{-7}$), *Veillonella* (LFC = -3.7, $q = 3.53 \times 10^{-6}$), *Actinomyces* (LFC = -4.03, $q = 3.98 \times 10^{-6}$) and *Gemella* (LFC = -1.16, $q = 6.13 \times 10^{-4}$).

### Differential analysis between noma and global healthy saliva samples indicate genera which are significantly different between healthy and diseased states

In order to further determine which genera were significantly different between the global healthy saliva dataset and the noma saliva samples, we ran a differential analysis of normalised count data using the DESeq2 model (Fig 2A) [34]. Given that the dataset contained 1883 genera, many with very low counts, we focused our analysis on the most prominent taxa. We filtered for genera with a mean relative abundance greater than 1% across all noma and healthy samples, resulting in a core set of 16 genera for comparison (S4 Fig).

Of these 16 genera we identified 11 genera that were differently abundant between the noma and healthy groups ($q$-value $< 0.05$). Four genera were significantly enriched in the noma samples (Fig 2C and 2E) including *Treponema* (LFC = +4.65, $q = 4.13 \times 10^{-23}$), *Porphyromonas* (LFC = +2.68, $q = 2.41 \times 10^{-10}$), *Bacteroides* (LFC = +2.04, $q = 2.21 \times 10^{-7}$) and *Selenomonas* (LFC = +1.02, $q = 2.62 \times 10^{-2}$). Seven genera were significantly depleted in the noma samples and

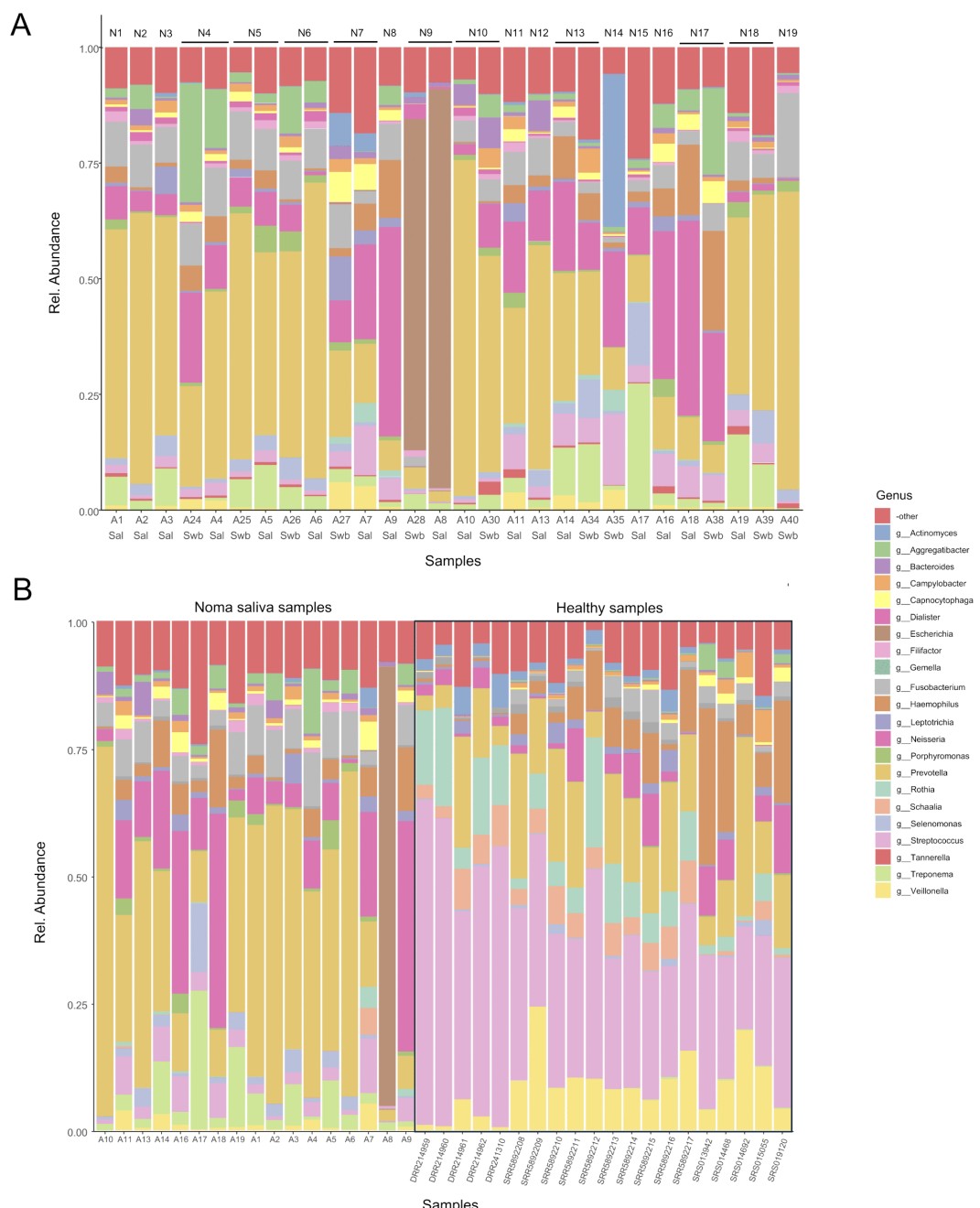

**Fig 1. Genus level comparison of relative abundance between noma samples and healthy controls.** The relative abundance of the top 22 most abundant genera is shown, with all other genera displayed under "-other". (A) Saliva and swab samples from noma participants. Participant IDs (N1-N19) are displayed on top and sample names (A1-A40) are displayed on the bottom, along with whether the sample came from saliva (Sal) or swab (Swb). (B) saliva samples from noma participants compared to a global dataset of healthy saliva samples (n = 20).

show significantly higher median counts in the global healthy saliva dataset (Fig 2B and 2D), these included *Streptococcus* (LFC = -4.55, $q$ = 2.64x10⁻³¹), *Veillonella* (LFC = -3.90, $q$ = 2.45x10⁻¹³), *Gemella* (LFC = -2.35, $q$ = 4.48x10⁻¹¹), *Schaalia* (LFC = -4.85, $q$ = 4.89x10⁻¹¹), *Rothia* (LFC = -5.61, $q$ = 9.61x10⁻¹¹), *Actinomyces* (LFC = -3.82, $q$ = 2.14x10⁻⁵), and

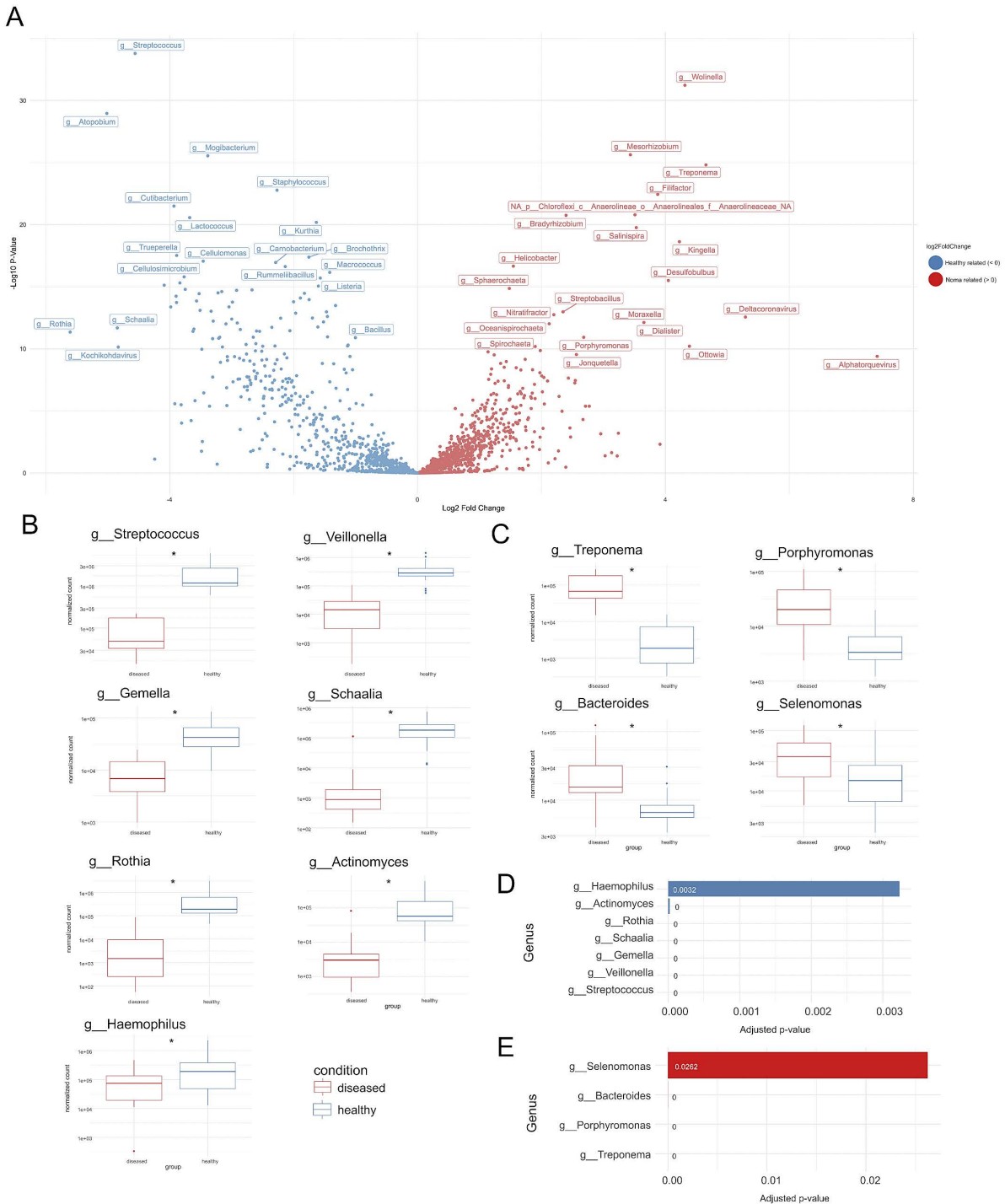

**Fig 2. Differential analysis between noma and healthy samples.** The Deseq model was used to calculate adjusted p-values and fold change for all genera. (A) A volcano plot with the log2 fold-change (healthy vs noma) across the x axis and the -log10 p-values across the y-axis, genera which have a positive fold change (> 0) are enriched in noma samples and genera which have a negative fold change (< 0) are enriched in healthy control samples. The normalized counts were plotted as boxplots for the genera with a mean relative abundance of more than 1% (top 1%) that are significantly related to healthy control (B) and noma samples (C). The boxplots summarise the distribution of data points, with the box representing the interquartile range (IQR), the horizontal line within the box representing the median and whiskers extending to 1.5 times the IQR, any outliers beyond this are shown as individual points. The adjusted p-values (q-values) are displayed as barplots for the top genera significantly related to healthy control (D) and noma samples (E).

*Haemophilus* (LFC = -1.92, $q$ = 3.2x10$^{-3}$). Some genera such as *Bacteroides* and *Veillonella* show a greater degree of variability as shown by the presence of outliers. The distribution of normalised counts between diseased and healthy groups for the 11 genera are visualized in Fig 2B and 2C. The adjusted p-values (q-values) are visualized in Fig 2D and 2E.

## Machine learning and multivariate statistical analyses indicate genera which contribute most to healthy or diseased states

To further determine which genera are most strongly associated with healthy and diseased states, we ran three independent additional analyses; random forests, ordination (PCoA), and hierarchical clustering/CLR transformation (Fig 3). A random forests classifier was used to determine how well categorical metadata variables (noma vs healthy) predict differences in the microbial communities. Each genus was treated as a predictor for the metadata classification. The importance of individual genera in distinguishing between the healthy and diseased states was assessed by those with the highest percent increase in mean squared error (%IncMSE) derived from the random forest model. From the top most abundant genera (S4 Fig), these included typically orally pathogenic *Treponema* (%IncMSE = 3.85), *Bacteriodes* (2.43), *Porphyromonas* (2.07) and orally commensal *Streptococcus,* (2.51) and *Rothia* (1.94) (Fig 3A). These genera had the greatest influence on the model prediction and therefore contribute the most to the distinction between noma and global healthy saliva dataset samples.

A cross validation of the dataset (see Methods) was shown to exhibit complete prediction with an area under the receiver operating characteristic (ROC) curve (AUC) of 1.0 (Fig 3B). This shows that there is a distinct separation between the microbiomes of global healthy saliva dataset and noma samples. This near-perfect classification reflects the profound magnitude of dysbiosis in noma, where the microbial community structure is fundamentally distinct from that of healthy individuals (as visualized in the non-overlapping clusters of the PCoA ordination shown in Fig 3D). As an AUC of 1.0 indicates such a clear distinction between healthy and diseased samples, a permutation test ($n$ = 1000) was conducted to evaluate the statistical significance of the observed classification accuracy (see Methods). The mean accuracy of the permuted models was significantly lower than the observed accuracy of the RF model (Fig 3C), providing strong evidence that the models performance is not due to random chance or model over-parameterisation. However, this distinct separation should be interpreted in the context of the study limitations; the comparison against a global healthy dataset, rather than geographically matched controls, likely maximises the distinguishability of the two groups by combining disease-specific signals with geographic microbiome variations.

Principal Coordinates Analysis (PCoA) with Bray-Curtis dissimilarity was used to display variation across samples in relation to disease state (Fig 3D). There was a distinct separation between noma samples and global healthy saliva dataset samples. A PERMANOVA analysis on the Bray-Curtis dissimilarity-matrix confirmed this ($R^2$ = 0.40, p = 0.001). The relative abundance of bacterial genera was subjected to CLR transformation and z-score standardization, then the top 20 most abundant genera were selected for analysis and visualization. Fig 3E shows the genus-level variation across samples ranging from a CLR z-score of -6 (lower abundance) to +6 (higher abundance), hierarchically clustered by the complete method and Euclidean distances. Notably, certain genera exhibited disease-state-specific enrichment patterns, some genera had a higher mean CLR z-score in noma samples such as *Treponema* (0.90), *Filifactor* (0.86), *Porphyromonas* (0.75), and *Bacteroides* (0.70), which reveals they are more abundant in noma samples. Other genera had a higher mean CLR z-score in global healthy saliva dataset samples such as *Streptococcus* (0.80), *Schaalia* (0.80), *Rothia* (0.75), and *Actinomyces* (0.72) showing they are more abundant in global healthy saliva dataset samples (Fig 3F). Taken together this reveals which genera contribute the most to a healthy oral microbiome or a microbiome associated with noma.

## Recovery of novel *Treponema sp*. MAGs from noma participant microbiome samples

We next sought to obtain MAGs from species associated by multivariate and machine learning analysis with disease status in noma metagenomes through a binning process. *Treponema* MAGs made up 25 of 139 high-quality MAGs identified

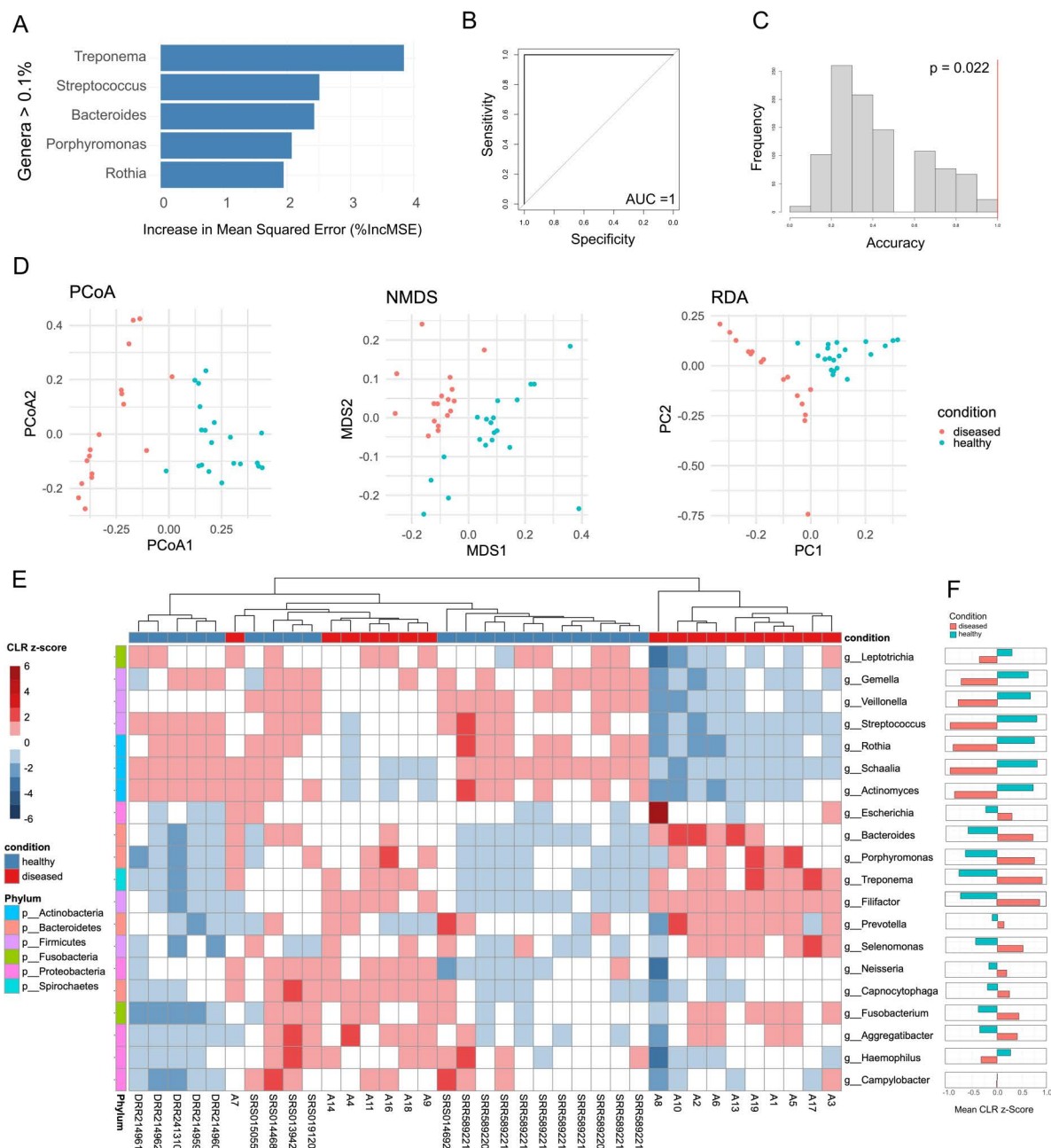

**Fig 3. Machine learning and multivariate statistical analyses.** Several analyses were applied to determine the differences between healthy and diseased states and which genera contributed the most to this difference. **(A)**. The genera with the highest percent increase in mean squared error (%IncMSE) showing they contributed most to the distinction between healthy and diseased in the random forest model **(B)** A receiver operating characteristic (ROC) curve from cross validation of random forest model **(C)** Null distribution of accuracy from permutation testing of the cross validation of the random forest model. This displays the classification accuracy across 1000 permutations, with the observed accuracy of the RF model against our data is shown as a red line. **(D)** Ordination of multivariate data into two dimensions using Principal Coordinate Analysis (PCoA) with Bray-Curtis dissimilarity. Points are coloured by disease state: healthy control (blue) and noma samples (red) **(E)** A heatmap displaying the z-score standardized centred log-ratio (CLR) of the top 20 most abundant genera, rows represent bacteria genera classified by Phylum and columns represent samples classified by disease state. A higher CLR z-score (Darker red) signifies higher abundance, and a lower score (darker blue) signifies lower abundance. The samples are hierarchically clustered by the complete method using Euclidean distances. **(F)** Mean CLR Z-scores by condition shown for each genus.

in total. Five high-quality *Bacteroides* MAGs were recovered, of which three could be identified at species level (two *Bacteroides heparinolyticus* and one *Bacteroides pyogenes*), while seven high-quality *Porphyromonas* MAGs were recovered (six of which were identified at species level: three *Porphyromonas gingivalis* and three *Porphyromonas endodontalis*).

When including medium-quality MAGs in further analysis targeting the most strongly disease associated genus, we recovered 40 unique medium- or high-quality *Treponema* MAGs (Fig 4). The MAGs belonged to 19 species, of which 14 were novel and could not be assigned to a known species in the genome taxonomy database (Fig 4A). The remaining five species, each of which only appeared in one MAG across all samples, were identifiable at species level against the genome taxonomy database, and were therefore not deemed novel: *Treponema lecithinolyticum*, *Treponema maltophilum*, *Treponema medium*, *Treponema* sp014334325, *Treponema* sp905372235. We positioned all recovered high-quality *Treponema sp.* MAGs in the context of previously characterised oral treponeme reference strains within NCBI RefSeq, as well as the agent of syphilis, *Treponema pallidum* subsp. *pallidum*. A maximum-likelihood phylogenetic tree constructed based on single nucleotide polymorphisms (SNPs) in soft core genes (Fig 4B) showed that novel *Treponema* species, as well as known species, were distributed across diverse clades with varying degrees of relatedness to oral reference strains, emphasizing the previously uncharacterized diversity of *Treponema* in the oral microbiome.

'*Treponema* sp. A' was the most prevalent novel *Treponema* species and was present as a medium- or high-quality MAG in samples from 11/19 participants (57.9%) (S5 Table). We observed marked genetic diversity within *Treponema* sp. A, in which the average nucleotide identity (ANI) between MAGs varied upwards of 96% (Fig 4C), indicating the presence of various strains. Three *Treponema* sp. A MAGs (Avii, Aviii, Axi) recovered from different participants appeared to be very closely related, with 99.8% ANI in each pairwise comparison. Five other novel *Treponema* species were present in samples from more than one noma participant, but none at a prevalence of 30% or more.

One high-quality MAG recovered from *Treponema* sp. A (Aix) contained a 926 bp partial 16S fragment which shared 99.46% identity with a partial 16S rRNA gene from uncultured *Treponema* identified within a subgingival plaque sample from a noma participant published in 2012 (GenBank Accession no. AM420013) [21]. We downloaded the 1505 bp AM420013 sequence to use as a 16S rRNA reference gene for *Treponema* sp. A, to screen across 16S rRNA genes recovered in our samples which could not be binned to a putative MAG. In addition to the eleven participants from which a medium- or high-quality *Treponema* sp. A MAG was recovered from swab or saliva samples, we identified a further four participants with partial *Treponema* sp. A 16S rRNA genes in their microbiomes. Through a combination of MAG and 16S rRNA gene recovery, we therefore detected *Treponema* sp. A in 15/19 noma participants assessed in this study, including 90% (8/9) of Stage 2 participants with oedema, and 100% (7/7) of Stage 3 participants with gangrenous lesions (S5 Table). *Treponema sp*. A was not detected in the participant with Stage 1 disease, nor the two participants with Stage 4 disease (scarring). Neither *Treponema* sp. A MAGs nor 16S rRNA genes were recoverable from the 20 metagenomes in the healthy global saliva dataset.

We examined previous metagenomic data sets for the presence of *Treponema* sp. A. Available 16S rRNA gene fragments (V1–3 region) from a study in Niger, in which samples from 12 healthy children, 12 children with Stage 1 noma (referred to as acute necrotizing ulcerative gingivitis in this study) and 12 children with established noma disease were collected [20], to determine the prevalence of *Treponema sp*. A within the different groups of their study by aligning 16S rRNA gene fragments to the AM420013 16S rRNA reference gene with BLAST. If 99% identity between 16S rRNA gene fragments was considered as a same species hit, *Treponema sp*. A was detected in 2/12 (16.7%), 10/12 (83.3%) and 7/12 (58.3%) of healthy, Stage 1 noma and established noma samples, respectively. Chi-square testing confirmed significant differences between the groups (p = 0.004). If the same-species threshold was dropped to 98%, all 12 samples from participants with stage 1 noma contained *Treponema sp*. A, while the prevalence within the healthy and established noma groups remained the same (2/12). However, sequences of the variable regions within 16S rRNA genes cannot provide the same level of taxonomic resolution as entire gene sequences, and sequence fragments with 97–99% identity should be treated with caution when inferring same species [35].

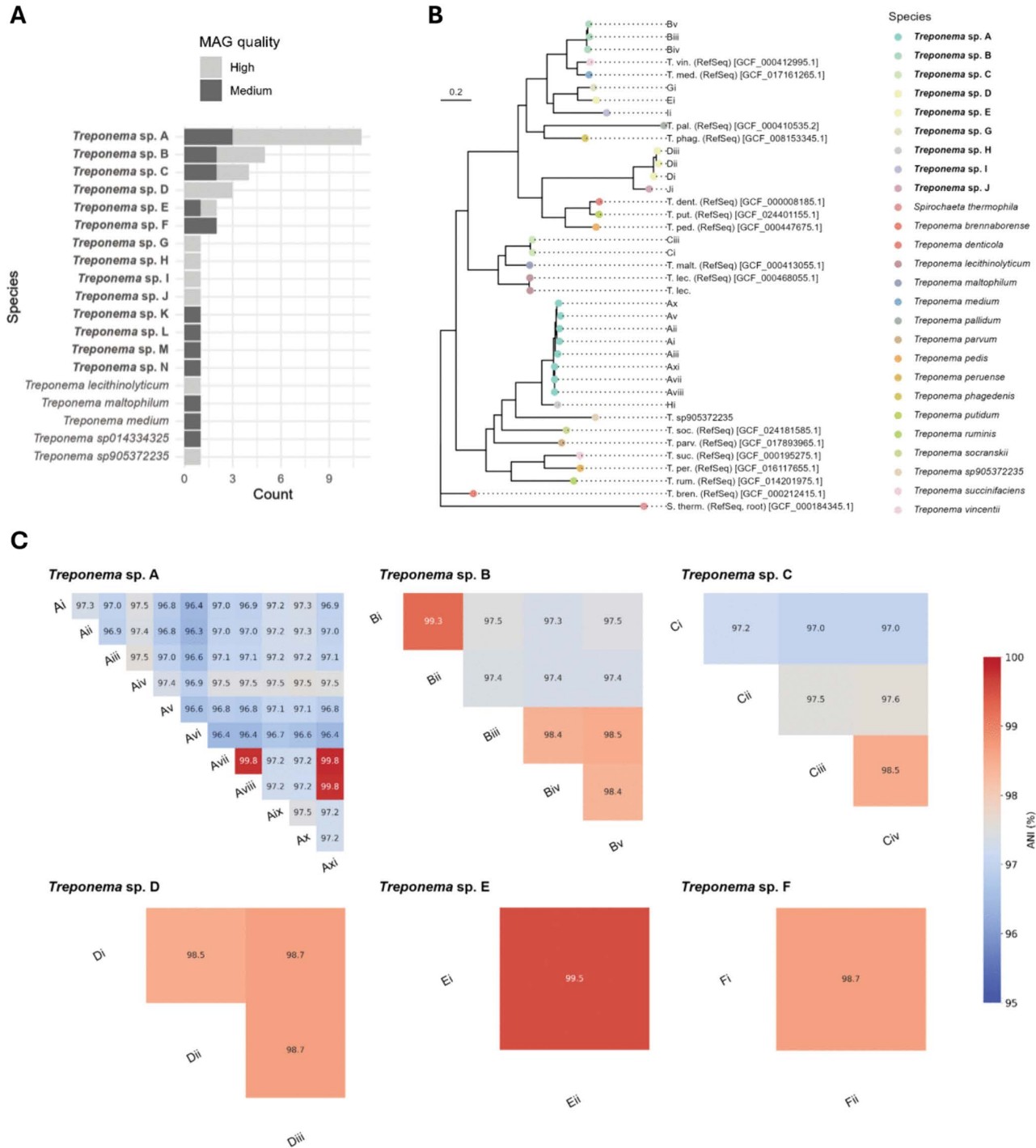

**Fig 4. Summary and taxonomic placement of Treponema MAGs recovered from metagenomes.** Novel species are in bold. a) frequency of different Treponema species MAGs; b) core-genome SNP maximum-likelihood tree placing high-quality Treponema MAGs within the context of oral Treponema species references; c) matrices showing ANI over aligning regions of MAGs within each Treponema species from which multiple MAGs were recovered.

## Antimicrobial resistance determinants identified in noma participant oral microbiomes

We identified 39 unique AMR determinants in noma oral microbiome samples (Fig 5A). Determinants putatively conferring resistance to beta-lactams and tetracyclines were the most prevalent, namely CfxA3 (17/19 participants) and Tet(Q) (18/19 participants), respectively. This pattern was also observed in the global dataset of saliva metagenomes from healthy individuals (Fig 5A). However, NimI metronidazole resistance determinants, which were not detected in healthy saliva metagenomes, were detected in 8/19 noma participant samples. Additionally, aminoglycoside resistance determinants were more commonly detected in noma samples, with *aph(3")-Ib* being identified in 12/19 (63.2%) of samples, while each aminoglycoside resistance determinant identified in samples from the healthy saliva dataset was not present in more two samples. In contrast, macrolide resistance determinants *mef*(A) and *msr*(D) were detected in all saliva metagenomes from healthy individuals, but only in samples from 3/19 and 1/19 participants with noma, respectively.

To identify the specific organisms harbouring antimicrobial resistance genes (ARGs) in noma samples, all medium- and high-quality MAGs were screened against the Resfinder database. Of the ARGs that were binned to MAGs, three of the four *cfxA3* genes and all four *nimI* genes were identified in putative *Prevotella* genomes (Fig 5B). These ARGs can confer resistance to two of the treatments routinely given to noma participants at our study hospital (co-amoxiclav and metronidazole). ARGs known to confer resistance to the gentamicin (aminoglycoside), the third antibiotic routinely given, could not be binned to medium- or high-quality MAGs. No known resistance determinants were identified in *Treponema* MAGs.

## Discussion

Noma, a devastating disease primarily affecting children in abject poverty, has long been understood to result from an opportunistic bacterial infection, evidenced by its prompt response to antibiotics [36–38]. However, the precise microbiological agents responsible for noma have remained unclear, hampering prevention, diagnosis, and treatment of the disease.

To interpret the shifts observed here, it is useful to note that the oral cavity is a complex ecological environment with multiple specific niches [39]. Generally, a healthy oral microbiome is dominated by typical abundant species, which are thought to be largely stable in humans globally [40–42] (as reflected the global healthy dataset, Fig 1) with complex immunological communication between the flora and the host. Factors such as the shared environment [43], diet, hygiene, and immunological status and infection have all been demonstrated to substantially affect the oral microbiome and in some cases result in periodontal disease [44]. Chronic malnourishment, poor oral hygiene, high rates of co-morbidities, and, in the case of children, immature immune systems, are all common factors applicable to noma participants [16,18]. However, not all children with these factors will develop noma, suggesting additional factors, such as presence or absence of specific microorganisms, amongst other things, are key to disease development.

Previous metagenomic 16S rRNA gene surveys noted a substantial loss of bacterial diversity in oral microbiomes of noma participants [22]. Similarly, our results reveal a drastic shift in microbial populations within the oral cavity of noma participants compared to that of global healthy saliva microbiomes (Fig 1). In particular, differential analysis showed that noma metagenomes were significantly enriched with *Treponema*, *Porphyromonas*, *Bacteroides,* and *Selenomonas*, whereas the global healthy saliva dataset comprised primarily of *Streptococcus*, *Veillonella*, *Gemella*, *Schaalia*, *Rothia*, *Actinomyces,* and *Haemophilus* species (Fig 2). The shift from a predominance of acid tolerant Gram-positive facultative anaerobes in healthy controls to mainly Gram-negative proteolytic obligate anaerobes, commonly associated with inflammation, mucosal invasion, and opportunistic infection, in the oral microbiomes of noma patients is notable. This microbial transition may suggest that oral salivary pH may be mildly alkaline in individuals with noma, favouring the growth of Gram-negative proteolytic anaerobic pathogens at the expense of Gram-positive facultative anaerobes [45]. Examining the oral pH of children with noma warrants inclusion in future research.

The presence and abundance of individual genera in isolation is not adequate to deconvolute complex microbial interactions in a polymicrobial disease [46]. To identify key microbial signatures in the oral microbiomes of noma participants

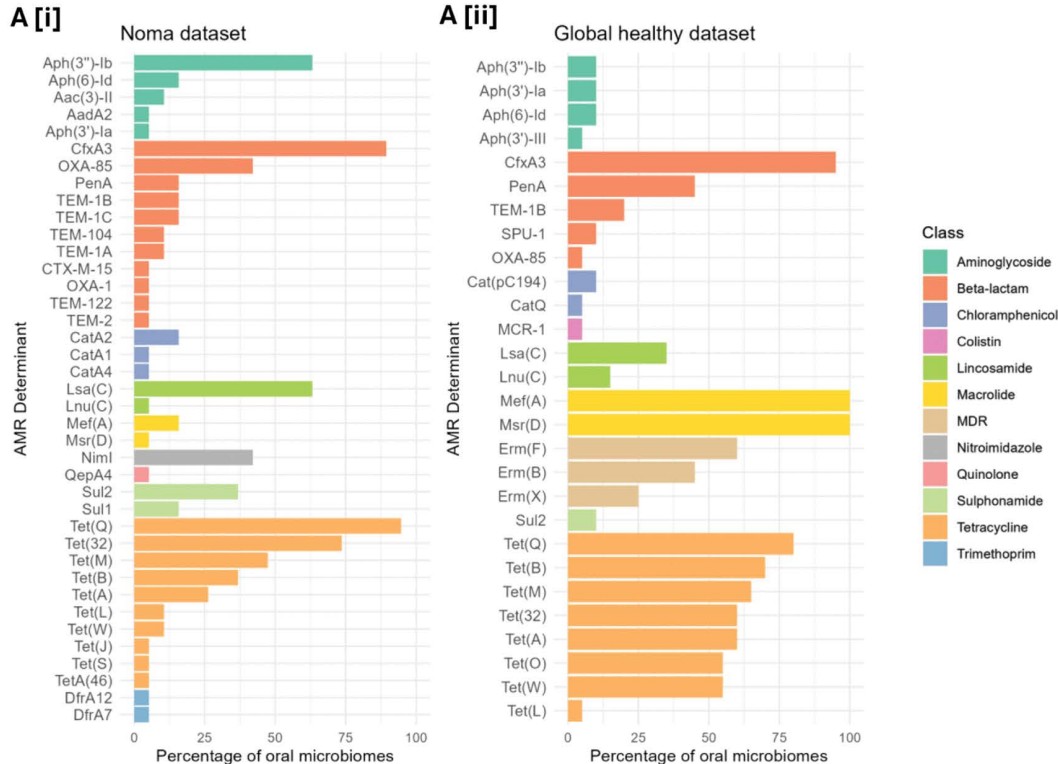

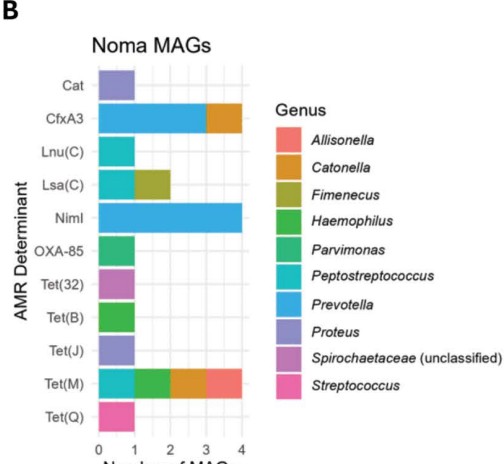

**Fig 5. Description of detected AMR determinants in noma metagenomes.** AMR determinants identified in: a) metagenome samples from [i] noma participants in this study, and [ii] saliva samples from the healthy global dataset; b) medium- and high-quality MAGs recovered from noma oral microbiome samples. In part (a), AMR determinants are grouped by class, with the most abundant determinants appearing at the top of each class. In part (b), colors indicate the genus of the MAG in which the AMR determinant was identified.

vs. microbiomes of healthy individuals which may be indicative of noma, we employed machine learning and multivariate statistical analyses, techniques which analyse the presence, abundance, and interactions of various microbes to determine patterns which correlate with specific disease states [47,48]. The key genera identified as signatures of noma

disease by these analyses were *Treponema*, *Porphyromonas* and *Bacteroides* (Fig 3). Previously *Porphyromonas*, a genus which contains well known oral pathogens such as *P. gingivalis*, and *P. endodontalis*, has been associated with noma, but more recent doubt was cast on their role in noma due to higher prevalence being observed in healthy vs. diseased participants in a 16S rRNA study in Nigeria [20]. *Bacteroides* is a genus which to date has not been associated with noma. Studies from the 1960s and 1980s which suggested *Bacteroides* as a potential contributor to noma were referencing *Prevotella* species prior to their reclassification as a distinct genus [38,49]. More recent noma metagenomic 16S rRNA analyses did not find members of the genus *Bacteroides* at greater quantities than healthy controls [20]. *Treponema* are normal inhabitants of the oral cavity, however, specific *Treponema* species are recognised as key oral pathogens [50,51]. Members of the genus are also the causative agents of many diseases, often affecting children in impoverished tropical populations and often causing necrotic lesions, for example, syphilis, pinta, yaws, and bejel, the latter two causing lesions in oral cavities [52,53]. Furthermore several *Treponema* species are associated with other multi-etiogical veterinary diseases with comparable disease progression in defined stages [54], including ovine contagious digital dermatitis [55] and bovine digital dermatitis [56].

Despite links to oral pathology, polymicrobial disease and conditions that disproportionately affect marginalised populations [57], the role of *Treponema* and other spirochaetes in noma has often been overlooked, even though they are frequently reported [14,16,20,21,26]. The exception to this is a study by Whiteson et al, [20] which performed a multivariate analyses on 16S rRNA metagenomes extracted from subgingival fluids collected from distinct oral healthy and lesion sites from noma participants in Niger. Their results demonstrated that *Treponema,* along with *Prevotella*, *Peptostreptococcus*, and *Sharpea*, were indicators of noma. Further studies on the same cohort of children provided mixed results regarding the association of spirochetes with noma. There are several reasons as to why *Treponema* may have been overlooked or proved inconsistent in previous microbiological and metagenomic surveys. *Treponema* are notoriously fastidious, and can often be only cultured on highly complex media in anaerobic conditions [38,58]. Furthermore, limitations of 'universal' 16S rRNA primers have been demonstrated to underestimate *Treponema* abundance and diversity in samples [22,59,60], thus their presence being likely substantially underestimated in previous surveys of noma.

Notably, we identified an uncultured, novel *Treponema* species (*Treponema* A) with high confidence from the majority (15/19) of noma patients in this study (Fig 4). Examination of the phylogeny of this novel species reveals it is more closely related to *Treponema socranskii*, a species frequently associated with severe periodontal tissue destruction [61], than it is to other known oral treponemes. Examination of previous 16S rRNA noma and associated ANG metagenomes also revealed the presence of *Treponema* sp. A in the majority of lesions sampled from participants with noma and ANG, while it was detected at very low frequency in the oral cavity of age matched control metagenomes in the same study [20]. The detection of this specific, novel treponeme in the oral metagenomes of noma participants from two independent studies, using different metagenomic techniques, and its apparent very low frequency in healthy oral microbiomes, suggests that *Treponema* A may be associated with the development of noma. Future, more comprehensive microbiome and genomic investigations will be important to confirm the robustness of this association and clarify its potential role in disease pathogenesis.

Whilst noting the overall shifts in the oral microbiomes of noma participants compared to healthy individuals, many previous studies on noma have limited the role of commensal bacteria to those of potential opportunistic pathogens, overlooking the potential protective role commensal bacteria may contribute to preventing noma. In many microbiotas, the maintenance of a commensal, normal microbiota is essential to the health function of the associated organ/tissue, as the commensals prevent pathogen dominance [62,63]. Notably, our analyses, in addition to the identification of the presence of *Treponema*, *Bacteroides* and *Porphyromonas*, also identified the lack of *Streptococcus* and *Rothia* species as a key indicator of noma disease (Fig 3).

Both *Streptococcus* and *Rothia* species are highly associated with healthy oral microbiomes in both adults and children [64,65]. *Streptococcus* spp. are one of the first colonisers of the oral cavity and are key in periodontal development and

maintaining oral health [65]. Similarly, *Rothia* species usually colonise the mouth after the age of one and are persistent colonisers for life [64]. Depletion of these species from oral microbiomes can be due to various factors, including poor oral hygiene, antibiotic use and compromised immune status, either through infection or chronic or acute malnourishment [44], all of which have been associated as risk factors for noma development [16,18]. The association of the depletion of these species as a key indicator of noma suggests that attempts to recolonise the oral cavity with *Streptococcus* and *Rothia*, or targeted prevention of their depletion altogether, may be a potential low cost, protective approach to reduce the incidence of noma. Multiple *Streptococcus* species, including *Streptococcus salivarius* and *Streptococcus mutans* are common food supplements, possess 'generally recognized as safe status', and are marketed as oral probiotics. Multiple studies have demonstrated health benefits of oral probiotics as adjunct therapies for treatment of dental caries and periodontitis [66,67], although overall robust clinical evidence of health benefits in the oral cavity remains limited [68,69]. Despite this, further investigation of oral probiotics as an adjunct strategy for noma prevention in at-risk children may be warranted to explore their potential protective role.

It is notable that *Prevotella,* a genus that has repeatedly linked to noma [16,20,21,25,38], whilst overall more abundant in noma metagenomes in this study (Fig 1), was not found to be significantly different compared to global healthy metagenomes (Figs 2 and S2), nor was *Prevotella* identified as a significant contributor to noma by machine learning or multivariate statistical analyses (Fig 3), suggesting that *Prevotella* may simply be more abundant due to poor oral hygiene or as a result of depletion of other microorganisms, but may not contribute directly to noma disease. Future, more comprehensive microbiome sequencing will be important to clarify whether this observed lack of significance of *Prevotella* relative to previous reports on noma microbiology reflects true biological differences.

The greater number of different ARGs identified in noma samples compared to the healthy global dataset (Fig 5) is perhaps a reflection of the altered taxonomic profiles of noma-associated oral microbiomes, although other local epidemiological factors not accounted for in our study may also contribute. Both beta-lactam and metronidazole resistance determinants, which were common in noma samples, were binned to putative *Prevotella* MAGs. Despite being abundant across noma participant microbiomes, *Prevotella* was not identified in our analysis as being a noma-associated genus (Fig 2). Therefore, the importance of identifying AMR determinants in *Prevotella* MAGs is debatable, as killing these organisms may not be clinically beneficial. While no AMR determinants could be linked to *Treponema* MAGs, the various novel *Treponema* species described here have not previously been cultured. Culture and antimicrobial susceptibility testing are vital next steps that may enable narrowed-spectrum therapy, including the possibility of oral-only antibiotics regimens for the treatment of noma if the novel noma-associated organism (*Treponema* sp. A) is proven to be pathogenic, and if phenotypic susceptibility data match genomic predictions.

The primary limitation of this preliminary study is focused on the use of a global healthy saliva dataset instead of generating healthy oral saliva microbiomes from a similar population to that of the noma participants. Since we found significant differences between countries (USA, Denmark and Japan) within the global healthy saliva dataset, the geography of the samples may be a confounding factor when comparing healthy and diseased samples. However, differential analysis showed the magnitude of difference between the noma and healthy datasets was substantially greater than between healthy datasets in several genera including *Treponema*, *Rothia*, and *Streptococcus* (S2 Fig). Despite these results and general acceptance that healthy oral microbiomes are stable globally [40], it is possible that our global dataset, originating from individuals in high-income countries in North America, Europe and Asia, may not be reflective of typical healthy African population oral microbiomes [70]. Ideally, future studies will include healthy controls from the same geographic location as the noma samples. Furthermore, since we only have 17 saliva samples from noma participants, the study is powered to detect only large effect sizes but may miss smaller but still biologically relevant differences. The metagenomic classification at the genus level, while necessary for specificity and reliability [71], may also miss species or strain-level variations. Despite these limitations, this data provides the first microbiological profiling of noma using deep shotgun metagenomics and highlights key shifts in microbial populations within the oral cavity of noma participants.

The lack of progress in elucidating noma microbiology, despite previous attempts to do so, has led some to question the value of understanding the etiology of the disease. Indeed, the disease was virtually eradicated from Europe and North America long before the discovery of antibiotics [13]. However, lack of detailed understanding of noma microbiology, in addition to the lack of understanding of many facets of noma pathogenesis [3], hinders targeted prevention efforts, hampers therapeutic choice, increases the risk of development of AMR, and limits the potential development of diagnostics, all of which are essential tools and considerations for eradicating noma more rapidly [72]. We are hopeful that the analysis presented here will provide a foundation for targeted microbiological studies of noma the future.

## Methods

### Ethics statement

The study protocol was reviewed and approved by the following ethical review boards: i) the Usman Danfodiyo University Teaching Hospital Health Research and Ethics Committee (UDUTH/HREC/2022/1152/V2), ii) the Sokoto State Ministry of Health Research Ethics Committee (SKHREC/042/2022), iii) the National Health Research and Ethics Committee in Nigeria (NHREC/01/01/2007-02/12/2022), iv) The Liverpool School of Tropical Medicine Ethical Review Board (22–045) and iv) the MSF Ethical Review Board (2326), which acted as primary sponsor.

### Participant recruitment

We conducted a pilot cross-sectional study at the Noma Children's Hospital, Sokoto, Nigeria from 16th October 2023–23rd February 2024, utilizing a questionnaire and collection of saliva and swab samples.

We enrolled pediatric participants admitted to the Noma Children's Hospital with noma through a non-probability convenience sampling. Inclusion criteria consisted of participants who arrived at the Noma Children's Hospital and were: i) diagnosed as having Stage 0 (at risk) to stage 4 noma; ii) aged below 12 years; and iii) whose caregiver provided consent and participant provided assent (where applicable). Exclusion criteria consisted of participants who arrived at the Noma Children's Hospital and were i) diagnosed as having Stage 5 noma or any other condition; ii) aged 12 years and above; and iii) whose caregiver did not provide consent or participant did not provide assent (where applicable). In addition to the pre-specified exclusion criteria in the protocol, the following were applied during recruitment at the hospital and not explicitly included in the original written protocol due to an oversight: iv) a recent history of consuming any antibiotics in the seven days prior to admission and v) having receiving any treatment (including wound debridement and dressing) in another hospital prior to admission to Noma Children's Hospital. These additional criteria were discussed and agreed upon by the research team prior to implementation to ensure that biological samples were obtained before any treatment interventions.

Formal written consent was obtained from the guardian of each child. Participants aged seven to 12 years provided written assent. Information sheets and the informed consent and assent forms were written in English and Hausa.

### Clinical information and questionnaire

Information routinely collected at the hospital included participant socio-demographic details, noma stage on admission, Mid Upper Arm Circumference (MUAC), weight, height, and vaccination history (mostly self-reported and subsequently confirmed or by vaccination card). Study specific data collected included caregiver demographics and recent treatment history.

### Sample collection and storage

Saliva Samples were collected using DNA/RNA Shield SafeCollect Saliva Collection Kit (Zymo Research, USA) and swabs were collected using a DNA/ RNA Shield Collection Tube with Swab (Zymo Research, USA), following

manufacturer's instructions. Buccal swab samples were collected by rubbing the swab against the inner diseased cheek six times. For saliva samples, participants were asked to dribble saliva into the sample tube until the required volume (1ml) had been obtained.

### DNA extraction and sequencing

Total DNA was extracted using the EchoLUTION Tissue DNA Micro Kit (BioEcho, Germany) as per manufacturer instructions. The metagenomic DNA sample was fragmented into short fragments. These DNA fragments were then end-polished, A-tailed, and ligated with full-length adapters for Illumina sequencing before further size selection. PCR amplification was then conducted unless specified as PCRfree. Purification was then conducted through the AMPure XP system (Beverly). The resulting library was assessed on the Agilent Fragment Analyzer System (Agilent) and quantified to 1.5nM through Qubit (Thermo Fisher Scientific) and qPCR. Whole metagenomic sequencing (paired-end 150 bp Illumina reads) was carried out by Novagene. The qualified libraries were pooled and sequenced on Illumina platforms, according to the effective library concentration and data amount required.

### Sequence processing

Raw sequence files were quality assessed using FASTQC (https://github.com/s-andrews/FastQC). Trimming with Trimmomatic [73] version 0.39 was performed to remove adaptor sequences and low-quality bases with a sliding window quality cutoff of Q20 and a minimum read length of 50 bp. Human (host) DNA was filtered out by aligning reads to the human genome sequence CRCh38.p13 using Bowtie2 (https://github.com/BenLangmead/bowtie2) Version 2.2.5 with the "-un-conc" option.

### Global healthy saliva sequence collection

Saliva sequences from healthy oral microbiomes were found by searching for metagenomic saliva samples that had been sequenced with Illumina technologies. We identified saliva metagenomes from 20 healthy individuals from three countries across three continents (USA, Japan and Denmark). The samples from Japan and Denmark (SRR and DRS) were downloaded from the ftp site on ENA using wget in December 2024. The samples from the USA (SRS) were part of the Human Microbiome Project [33] and downloaded directly from https://www.hmpdacc.org/HMASM/ (NCBI BioProject PRJNA43017). The reads were quality-checked with FASTQC.

### Taxonomic classification and relative abundance

Read based taxonomic assignment was run against the filtered reads using Kraken2 v2.1.3 [74,75] and Bracken v2.9 [76]. The bracken reports were converted to a metaphlan format and combined using KrakenTools scripts (https://github.com/jenniferlu717/KrakenTools) [77]. This combined report was imported into R v4.3.1 as a tree summarized experiment object with the mia v1.10.0 package [78]. An additional assay was constructed for relative abundance and an alternative experiment (altExp) was constructed at the genus level using the SingleCellExperiment v1.24.0 package. The top 20 most abundant genera at each taxonomic level were selected from across the entire dataset, with all others designated "-other", and visualised with the plotAbundance function of miaViz v1.10.0.

### Differential analysis of genera

Differential analysis used the DESeq2 model [34] as part of the DESeq2 v1.42.1 package. The statistical model was defined by the design formula "~ condition", which tested for the effect of the disease state ('diseased' vs. 'healthy') on the normalized counts of each genus individually. The adjusted p-values and log2 fold change (LFC) for all genera were plotted together in a volcano plot. A genus was considered significantly differentially abundant if the False Discovery Rate

(FDR) adjusted p-value (padj) was less than 0.05. The most abundant genera (top 1%) were determined across the entire dataset and those which were significantly related to the healthy saliva dataset and noma samples had their normalized counts plotted as boxplots. Subject ID was not included as a variable in the model as the analysis was based on a cross-sectional design comparing two independent groups of subjects.

## Multivariate statistical analyses and machine learning approaches

The tree summarized experiment object was converted into a Phyloseq object with the mia package (v1.10.0) for the random forests and ordination analyses. Random forest classifiers were applied using the randomForest package (v4.7-1.2) in R (v4.3.1) based on 500 estimator trees to identify key predictors of microbial community composition and their association with noma or healthy states. This machine learning technique was utilised to rank variable importance, expressed as percent increase in mean squared error (%IncMSE) derived from the random forest model, those variables with the highest %IncMSE contributed the most to the metadata classification. Each genus was treated as a predictor for the metadata classification of disease state (noma vs healthy). A cross validation of the dataset taking 20% of samples as test data and 80% as training data, was used to assess the predictive power of the classifier. To prevent data leakage, the test subset was isolated from the training phase and was utilised exclusively for final model evaluation. Predictive power was assessed as an area under the receiver operating characteristic (ROC) curve (AUC) between 0 and 1 using the pROC package (v1.18.5). A permutation test ($n = 1000$) was conducted to evaluate the statistical significance of the observed classification accuracy. This allowed for a null distribution of accuracy to be plotted. The observed accuracy of the random forest model was overlayed on the null distribution.

Ordination techniques including Principal Coordinate Analysis (PCoA) were performed on a Bray-Curtis dissimilarity matrix of the data with the ordinate and distance functions from the phyloseq package (v1.46.0). The first two axes (i.e., PCoA1 and PCoA2) were visualized using the ggplot2 package (v 3.5.1), where samples were represented as points colored by their disease state.

Using the mia package (v1.10.0), a centered log-ratio (CLR) transformation was run on the relative abundance of genera, with a pseudocount to account for compositional effects. A z-score standardization was then applied across genera to normalize variability. The top 20 most abundant genera were selected and a heatmap was generated with the pheatmap package (v 1.0.12) annotated with disease state. Mean CLR z-scores were calculated for each genus in healthy and diseased conditions by aggregating the data by disease state and summarizing with the dplyr package (v1.1.4).

The vegan package (v2.6-8) was used to run PERMANOVA across the entire dataset for age, sex, sample type and disease status, and to run ANOSIM for respondent ID. For PERMANOVA the proportion of variance explained by each variable was reported as the $R^2$ effect size, for ANOSIM the R statistic was used. All tests used a Bray-Curtis dissimilarity matrix calculated with the phyloseq package (v1.46.0). For the healthy dataset differences in country of origin, individual genera were assessed using the Kruskal-Wallis test. The Wilcoxon rank sum test was used to compare the relative abundance of certain genera within the noma dataset (swab vs saliva) and between the noma and global healthy saliva dataset samples (noma vs healthy). The p-values were adjusted for multiple hypothesis testing using the Benjamini-Hochberg method to control the false discovery rate FDR. An FDR-adjusted p-value, or q-value, of less than 0.05 was considered statistically significant.

## Metagenome assembly and MAG recovery

Reads were assembled using metaSPAdes [79] version 3.11.1 at default settings. QUAST [80] version: 5.0.2 was used to check the quality of metagenome assemblies, with assembly statistics generated using contiguous sequences (contigs) of ≥ 500 bp in length. Contigs were indexed using the Burrows Wheeler Aligner (BWA) Version 0.7.17 before the raw reads were mapped against the indexed contigs using "bwa-mem" (https://github.com/lh3/bwa) [81] The contigs and resulting sorted BAM files were parsed to the "jgi_summarize_bam_contig_depth" script from MetaBAT2 [82] version 2.17,

before the resulting depth files were used by MetaBAT2 to bin assembled contigs of ≥ 2,500 bp in length to metagenome assembled genomes (MAGs). CheckM [83] version 1.1.2 was used to assess the completeness and contamination of each MAG. CheckM outputs were compared to standards set by the genome standards consortium to classify MAGs as high-quality (>90% completeness, <5% contamination), medium-quality MAGs (>50% completeness, <10% contamination) or low-quality (<50% complete and <10% contamination, or >10% contamination) [84]. MAGs classified as medium- or high-quality progressed to the next analysis steps.

## MAG taxonomic profiling and placement of *Treponema* MAGs

MAGs underwent taxonomic classification using GTDB-Tk using with the "classify" workflow version 2.1.1 against the Genome Taxonomy Database Version 09-RS220 (24th April 2024) [85]. The JSpeciesWS ANIm tool was used for pairwise comparison of *Treponema* MAGs which could not be assigned to a known species by GTDB-Tk, to identify novel species groups [86,87] In the case that the same MAG was recovered from swab and saliva samples from one participant, the MAG with the greatest CheckM completeness proceeded to next steps of analysis.

## *Treponema* MAGs in phylogenetic context

Published RefSeq genomes for known *Treponema* species were downloaded, including *Treponema brennaborense* (GCF_000212415.1), *Treponema denticola* (GCF_000008185.1), *Treponema lecithinolyticum* (GCF_000468055.1), *Treponema maltophilum* (GCF_000413055.1), *Treponema medium* (GCF_017161265.1), *Treponema pallidum* subsp. pallidum (GCF_000410535.2), *Treponema parvum* (GCF_017893965.1), *Treponema pedis* (GCF_000447675.1), *Treponema peruense* (GCF_016117655.1), *Treponema phagedenis* (GCF_008153345.1), *Treponema putidum* (GCF_024401155.1), *Treponema ruminis* (GCF_014201975.1), *Treponema socranskii* subsp. *buccale* (GCF_024181585.1), *Treponema succinifaciens* (GCF_000195275.1), and *Treponema vincentii* (GCF_000412995.1). Additionally, the RefSeq *Spirochaeta thermophila* (GCF_000184345.1) genome was downloaded to act as an outgroup reference genome from a different genus within the Spirochaetaceae family. Annotations of our recovered *Treponema* spp. MAGs, as well as RefSeq genomes, were performed with Bakta Version 1.9.3 (full database Version 5.1) [88]. Panaroo [89] version 1.3.2 was used to retrieve soft core gene alignments to infer phylogeny, with clean mode set to moderate, a 70% sequence identity threshold for orthologue detection and a 90% threshold to ensure that genes included were present in at least 90% of genomes. Variant sites containing single nucleotide polymorphisms (SNPs) were then extracted from the alignments using SNP-Sites (https://github.com/sanger-pathogens/snp-sites) version 2.5.1. A maximum-likelihood tree was built using 1,000 bootstraps in IQ-TREE [90] version 1.6.1, with the most appropriate model (GTR + F + ASC + R3) determined by ModelFinder [91] within IQ-TREE. The tree was rooted by the *Spirochaeta thermophila* outgroup. The resulting treefile was visualized in the *ggtree* Version 3.12.0 package in R.

## 16S rRNA gene identification and reconstruction

To enable comparison to 16S microbiome datasets, Barrnap (https://github.com/tseemann/barrnap) Version 0.9 was used to quickly identify complete and partial ribosomal RNA genes in MAGs, where present. 16S rRNA genes of interest were queried against the GenBank nucleotide database using the Nucleotide Basic Local Alignment Search Tool (BLAST) webtool (megablast algorithm) to identify similar sequences of interest. However, 16S rRNA genes are often missing from MAGs due to difficulties both with resolving conserved regions in initial assembly algorithms and with handling resulting sequences in binning algorithms [92]. Therefore, after linking any given novel species MAG to a 16S rRNA gene sequence (in the minority situation where a 16S rRNA gene could be binned to the MAG), we were able to use the 16S sequence as a reference for the novel species. MATAM [93] was used alongside the SILVA SSU database Version 138.2 (https://www.arb-silva.de/documentation/release-1382/) to re-construct 16S rRNA genes in metagenome samples. We were then able

to confirm the presence of a novel species within samples from which a medium- to high-quality MAG from that species could not be recovered by querying MATAM assemblies against the MAG-linked 16S reference.

### Identification of antimicrobial resistance determinants

Metagenome assemblies from both noma microbiome samples and the global healthy saliva dataset, and MAGs, were screened against the ResFinder database using ABRicate (https://github.com/tseemann/abricate) Version 1.0.1, to identify putative antimicrobial resistance genes (ARGs). Thresholds for minimum nucleotide identity and minimum query cover were set at 90%.

## Supporting information

**S1 Table. Symptoms, clinical assessments, and treatment.**
(DOCX)

**S2 Table. Participant demographic information.**
(DOCX)

**S3 Table. Health facility visit and treatment.**
(DOCX)

**S4 Table. Sequencing and assembly statistics.**
(DOCX)

**S5 Table. Identification of _Treponema sp_. A in noma participant oral microbiome samples in MAGs and 16S rRNA genes.**
(DOCX)

**S6 Table. Healthy control metadata descriptions.**
(DOCX)

**S1 Fig. Counts of _Treponema_ across noma stages.**
(DOCX)

**S2 Fig. Counts of twelve genera across the noma dataset and the three separate healthy cohorts.**
(DOCX)

**S3 Fig. Statistical differences in relative abundance of genera in healthy control and noma samples of top 20 genera.**
(DOCX)

**S4 Fig. Top 1% of genera across entire dataset.**
(DOCX)

**S1 Data. Accessions for MAGs defined in this project.**
(CSV)

## Acknowledgments

We gratefully acknowledge the children and their families who participated in this study. We would like to thank Simon Wagstaff and Andrew Bennett at the LSTM Scientific Computing Unit for providing access to a high-performance computing server for data storage and analysis.

## Author contributions

**Conceptualization:** Michael Olaleye, Miriam Peters, Shafi'u Isah, Adeniyi Semiyu Adetunji, Elise Farley, Mark Sherlock, Adam P. Roberts, Mohana Amirtharajah, Stuart Ainsworth.

**Data curation:** Michael Olaleye, Angus M. O'Ferrall, Richard N. Goodman.

**Formal analysis:** Michael Olaleye, Angus M. O'Ferrall, Richard N. Goodman, Gregoire Falq, Nichoas J Evans, Stuart Ainsworth.

**Investigation:** Angus M. O'Ferrall, Richard N. Goodman.

**Methodology:** Angus M. O'Ferrall, Richard N. Goodman.

**Project administration:** Michael Olaleye, Shafi'u Isah, Adeniyi Semiyu Adetunji, Elise Farley, Mark Sherlock, Adam P. Roberts, Mohana Amirtharajah, Stuart Ainsworth.

**Resources:** Joseph Samuel, Donal Doyle, Diana Gomez, Gbemisola Oloruntuyi, Adam P. Roberts, Mohana Amirtharajah.

**Supervision:** Adam P. Roberts, Mohana Amirtharajah, Stuart Ainsworth.

**Visualization:** Angus M. O'Ferrall, Richard N. Goodman.

**Writing – original draft:** Michael Olaleye, Angus M. O'Ferrall, Richard N. Goodman, Adam P. Roberts, Stuart Ainsworth.

**Writing – review & editing:** Michael Olaleye, Angus M. O'Ferrall, Richard N. Goodman, Deogracia Wa Kabila, Miriam Peters, Gregoire Falq, Joseph Samuel, Donal Doyle, Diana Gomez, Gbemisola Oloruntuyi, Shafi'u Isah, Adeniyi Semiyu Adetunji, Elise Farley, Nichoas J Evans, Mark Sherlock, Adam P. Roberts, Mohana Amirtharajah, Stuart Ainsworth.

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
