## [Decision Letter · Decision Letter 0]

28 Jan 2026

Shotgun metagenomic analysis of the oral microbiomes of children with noma

Dear Dr. Ainsworth,

Thank you for submitting your manuscript to PLOS Neglected Tropical Diseases. After careful consideration, we feel that it has merit but does not fully meet PLOS Neglected Tropical Diseases's publication criteria as it currently stands. Therefore, we invite you to submit a revised version of the manuscript that addresses the points raised during the review process.

Please submit your revised manuscript within by Mar 29 2026 11:59PM. If you will need more time than this to complete your revisions, please reply to this message or contact the journal office at plosntds@plos.org. Please include the following items when submitting your revised manuscript:

We look forward to receiving your revised manuscript.

Kind regards,

Michael Marks

Academic Editor

Stuart Blacksell

Section Editor

Shaden Kamhawi

co-Editor-in-Chief

Paul Brindley

co-Editor-in-Chief

**Journal Requirements:**

At this stage, the following Authors/Authors require contributions: Michael Olaleye, Angus M. O'Ferrall, Richard N. Goodman, Deogracia Wa Kabila, Miriam Peters, Gregoire Falq, Joseph Samuel, Donal Doyle, Diana Gomez, Gbemisola Oloruntuyi, Shafi'u Isah, Adeniyi Semiyu Adetunji, Elise Farley, Nichoas J Evans, Mark Sherlock, Adam P Roberts, Mohana Amirtharajah, and Stuart Robert Ainsworth. Please ensure that the full contributions of each author are acknowledged in the "Add/Edit/Remove Authors" section of our submission form.

**Reviewers' Comments:**

Reviewer's Responses to Questions

**Key Review Criteria Required for Acceptance?**

**Methods**

-Are the objectives of the study clearly articulated with a clear testable hypothesis stated?

-Is the study design appropriate to address the stated objectives?

-Is the population clearly described and appropriate for the hypothesis being tested?

-Is the sample size sufficient to ensure adequate power to address the hypothesis being tested?

-Were correct statistical analysis used to support conclusions?

-Are there concerns about ethical or regulatory requirements being met?

Reviewer #1: The global healthy saliva dataset has been used as the comparator but the differences in geography, diet, socio-economic status, oral hygiene, and environmental exposures may confound interpretation. Please mention this methods. .

Reviewer #2: -Yes, objectives were clearly articulated.

-Yes, study design is appropriate, howevere could be better by supplementing bacterial culture and in vitro infection study.

-Yes, its clearly described.

-No

-No, for involved sample numbers. Yes for the bioinformatics analysis done to show data.

-No

**Results**

-Does the analysis presented match the analysis plan?

-Are the results clearly and completely presented?

-Are the figures (Tables, Images) of sufficient quality for clarity?

Reviewer #1: The functional analysis from metagenomics data (e.g. metabolic pathways or virulence factors) would have strengthen the study.

Results related to stage-specific microbial patterns are limited and sometimes difficult to interpret due to small numbers. Please see this part. Also, the random forest model shows near-perfect classification (AUC = 1.0), which is striking. While technically valid, this result warrants clearer contextualisation.

Reviewer #2: -Yes

-Yes

-Yes

**Conclusions**

-Are the conclusions supported by the data presented?

-Are the limitations of analysis clearly described?

-Do the authors discuss how these data can be helpful to advance our understanding of the topic under study?

-Is public health relevance addressed?

Reviewer #1: (No Response)

Reviewer #2: -Yes

-Yes

-Yes but can be improoved.

-Yes

**Editorial and Data Presentation Modifications?**

Reviewer #1: (No Response)

Reviewer #2: Minor revision

**Summary and General Comments**

Reviewer #1: This manuscript presents the first deep shotgun metagenomic analysis of oral microbiomes from children with acute noma. The identification of novel Treponema species such as Treponema sp. A, and the analysis of AMR determinants are notable strengths of the study.

Reviewer #2: Noma is a necrotizing disease that primarily affects malnourished and immunocompromised children. It is believed to result from a polymicrobial infection, and no single definitive causative agent has been identified to date. The manuscript by Olaleye et al. presents compelling and novel findings that suggest a probable causative organism—Treponema sp. A—identified from the complex oral microbiome of children at various stages of Noma. The authors also provide detailed information regarding sample inclusion/exclusion criteria, sample handling, storage, and analytical methods. However, the study relies heavily on metagenomic analysis using shotgun sequencing, a technique that carries inherent limitations and biases. To strengthen the impact and accuracy of the findings, further validation through isolation and culture of Treponema sp. A along with in vitro infection experiments would be ideal. Additionally, the relatively small sample size (across different Noma stages, age ranges, and genders) limits the strength of the conclusions.I have a few minor comments:

1. Lines 62–64: The sentence is unclear and should be rewritten for clarity.

2. Figure 1A & 1B: Fusobacterium abundance appears higher compared to the global dataset, yet this difference is not reflected in other analyses (Fig. 2). Please include differential abundance data for Fusobacterium in Figure 2.

3. Lines 403–413: Several analyses are referenced, but the corresponding table or figure is missing.

4. Please apply the Oxford comma consistently throughout the manuscript.

5. Lines 506–508: This sentence is confusing and should be rephrased for clarity.

6. The Discussion section is lengthy and contains extensive literature review, which distracts from the main findings. Removing unnecessary background information would improve the focus and readability.

7. Line 652: Please include the name of the company providing the kit.

PLOS authors have the option to publish the peer review history of their article (what does this mean? ). If published, this will include your full peer review and any attached files.). If published, this will include your full peer review and any attached files.

**Do you want your identity to be public for this peer review?** For information about this choice, including consent withdrawal, please see our For information about this choice, including consent withdrawal, please see our Privacy Policy ..

Reviewer #1: No

Reviewer #2: **Yes:** Deepak ChouhanDeepak Chouhan

**Figure resubmission:**
---

## [Decision Letter · Decision Letter 1]

5 Mar 2026

Dear Dr Ainsworth,

We are pleased to inform you that your manuscript 'Shotgun metagenomic analysis of the oral microbiomes of children with noma' has been provisionally accepted for publication in PLOS Neglected Tropical Diseases.

Best regards,

Michael Marks

Academic Editor

Stuart Blacksell

Section Editor

Shaden Kamhawi

co-Editor-in-Chief

Paul Brindley

co-Editor-in-Chief

Reviewer's Responses to Questions

**Key Review Criteria Required for Acceptance?**

**Methods**

-Are the objectives of the study clearly articulated with a clear testable hypothesis stated?

-Is the study design appropriate to address the stated objectives?

-Is the population clearly described and appropriate for the hypothesis being tested?

-Is the sample size sufficient to ensure adequate power to address the hypothesis being tested?

-Were correct statistical analysis used to support conclusions?

-Are there concerns about ethical or regulatory requirements being met?

Reviewer #1: The revised Ms is much approved which can be accepted for publication.

Reviewer #2: (No Response)

**Results**

-Does the analysis presented match the analysis plan?

-Are the results clearly and completely presented?

-Are the figures (Tables, Images) of sufficient quality for clarity?

Reviewer #1: The revised Ms is much approved which can be accepted for publication.

Reviewer #2: (No Response)

**Conclusions**

-Are the conclusions supported by the data presented?

-Are the limitations of analysis clearly described?

-Do the authors discuss how these data can be helpful to advance our understanding of the topic under study?

-Is public health relevance addressed?

Reviewer #1: The revised Ms is much approved which can be accepted for publication.

Reviewer #2: (No Response)

**Editorial and Data Presentation Modifications?**

Reviewer #1: (No Response)

Reviewer #2: (No Response)

**Summary and General Comments**

Reviewer #1: The revised Ms is much approved which can be accepted for publication.

Reviewer #2: (No Response)

PLOS authors have the option to publish the peer review history of their article (what does this mean? ). If published, this will include your full peer review and any attached files.). If published, this will include your full peer review and any attached files.

**Do you want your identity to be public for this peer review?** For information about this choice, including consent withdrawal, please see our For information about this choice, including consent withdrawal, please see our Privacy Policy ..

Reviewer #1: **Yes:** Binod RayamajheeBinod Rayamajhee

Reviewer #2: **Yes:** Deepak ChouhanDeepak Chouhan

---

## [Editor Report · Acceptance letter]

Dear Dr Ainsworth,

We are delighted to inform you that your manuscript, "Shotgun metagenomic analysis of the oral microbiomes of children with noma," has been formally accepted for publication in PLOS Neglected Tropical Diseases.

Best regards,

Shaden Kamhawi

co-Editor-in-Chief

Paul Brindley

co-Editor-in-Chief
